# The Elk-3 target Abhd10 ameliorates hepatotoxic injury and fibrosis in alcoholic liver disease

Tian-Zhu Li [1✉], Chun-Ying Bai[1], Bao Wu[2], Cong-Ying Zhang[3], SIRIGULENG[4]*, Wen-Tao Wang[5], Tie-Wei Shi[1] & Jing Zhou[1]

Alcoholic liver disease (ALD) and other forms of chronic hepatotoxic injury can lead to transforming growth factor β1 (TGFβ1)-induced hepatic fibrosis and compromised liver function, underscoring the need to develop novel treatments for these conditions. Herein, our analyses of liver tissue samples from severe alcoholic hepatitis (SAH) patients and two murine models of ALD reveals that the ALD phenotype was associated with upregulation of the transcription factor ETS domain-containing protein (ELK-3) and ELK-3 signaling activity coupled with downregulation of α/β hydrolase domain containing 10 (ABHD10) and upregulation of deactivating S-palmitoylation of the antioxidant protein Peroxiredoxin 5 (PRDX5). In vitro, we further demonstrate that ELK-3 can directly bind to the ABHD10 promoter to inhibit its transactivation. TGFβ1 and epidermal growth factor (EGF) signaling induce ABHD10 downregulation and PRDX5 S-palmitoylation via ELK-3. This ELK-3-mediated ABHD10 downregulation drives oxidative stress and disrupts mature hepatocyte function via enhancing S-palmitoylation of PRDX5's Cys100 residue. In vivo, ectopic Abhd10 overexpression ameliorates liver damage in ALD model mice. Overall, these data suggest that the therapeutic targeting of the ABHD10-PRDX5 axis may represent a viable approach to treating ALD and other forms of hepatotoxicity.

[1] Department of Molecular Biology, College of Basic Medical Science, Chifeng University, Chifeng 024000, China. [2] Department of Tissue and Embryology, College of Basic Medical Science, Chifeng University, Chifeng 024000, China. [3] Department of Pharmacy, College of Basic Medical Science, Chifeng University, Chifeng 024000, China. [4] Department of Physiology, College of Basic Medical Science, Chifeng University, Chifeng 024000, China. [5] Department of Pathogenic Biology, College of Basic Medical Science, Chifeng University, Chifeng 024000, China. ✉email: litianzhu@cfxy.edu.cn

Hepatotoxic injury results from injury to hepatocytes through alcoholic liver disease (ALD), infection (e.g., hepatitis B and C), or non-alcoholic steatohepatitis. Chronic hepatotoxic injury can lead to hepatic fibrosis–a serious, progressive condition that arises from the aberrant activation of fibrotic regenerative processes. Hepatic fibrosis involves the extensive deposition of type I collagen and other extracellular matrix components within the liver, inducing the development of scar tissue and compromise of liver function[1]. Hepatotoxic injury and fibrosis has been linked to the epithelial-mesenchymal transition (EMT) in hepatocytes, a de-differentiating process characterized by a loss of epithelial-like cellular characteristics[2,3]. The fibrogenic cytokine transforming growth factor β1 (TGFβ1) drives EMT induction and is thought to be an important regulator of hepatotoxic injury and fibrosis[4]. TGFβ1 can bind to both the type I and type II receptors (TGFβRI and TGFβRII) to induce serine/threonine kinase activity and downstream signaling[3]. EMT induction in hepatocytes has been detected in TGFβ1-treated primary hepatocytes[5] as well as primary hepatocytes derived from a CCl4-induced model of liver cirrhosis[6]. Notably, TGFβ1-driven hepatocyte EMT induction has been associated with ETS domain-containing protein (ELK-3) upregulation, with similar upregulation also being evident in cirrhotic tissue samples from human patients and in the CCl4-induced model system[7]. ELK-3 silencing and RAS-ELK-3 pathway suppression further inhibit EMT marker gene expression, highlighting ELK-3 as a key regulator of TGFβ1-driven hepatotoxic injury and fibrosis[7].

To date, the downstream mechanism(s) whereby TGFβ1-driven ELK-3 signaling regulates hepatotoxic injury and fibrosis remain unclear. Our preliminary analyses revealed a conserved ELK-3 binding site within the promoter region of α/β hydrolase domain containing 10 (ABHD10), a redox homeostasis-related gene which is highly expressed in hepatocytes[8–10]. Herein, we conducted bioinformatics analyses in severe alcoholic hepatitis (SAH) and healthy control liver transcriptomes to identify gene modules and genes associated with alcohol-induced hepatotoxic injury and fibrosis. We found that liver tissue samples from SAH patients and two murine models of ALD exhibit upregulation of ELK-3 expression and ELK-3 signaling activity coupled with downregulation of ABHD10 and upregulation of deactivating S-palmitoylation of the antioxidant protein Peroxiredoxin 5 (PRDX5). In vitro, we further demonstrate that ELK-3 can directly bind to the ABHD10 promoter to inhibit its transactivation. We show that TGFβ1 and epidermal growth factor (EGF) signaling downregulate hepatocyte ABHD10 via ELK-3. We provide evidence that ABHD10 inhibits oxidative stress and promotes mature hepatocyte function by downregulating S-palmitoylated PRDX5. In vivo, we demonstrate that ectopic Abhd10 overexpression ameliorates liver damage in ALD model mice. As such, the therapeutic targeting of the ABHD10-PRDX5 axis may represent a viable approach to treating ALD and other forms of hepatotoxicity.

## Results

### Bioinformatics analysis identifies *ABHD10* as a downregulated gene in SAH patients.

Assessing gene co-expression patterns can identify biologically-relevant functional connections between genes in disease states. Therefore, we conducted a co-expression module identification analysis of liver tissue transcriptomes from SAH patients (Maddrey's discriminant function >32) ($n = 15$) versus those of healthy controls ($n = 7$) (GEO acc. no.: GSE28619)[11] in order to identify gene modules associated with hepatic fibrosis. This analysis revealed 15 gene modules that were significantly enriched for SAH (adj. *p* value <0.05; Fig. 1a and Supp. Table S4). M3, M4, and M7 were the largest of these 15 modules (Fig. 1b).

We selected the largest module among these–M3—for further investigation. M3 member genes were subjected to Reactome gene-set enrichment analysis. M3 was significantly enriched for amino acid metabolism, steroid metabolism, glyoxylate metabolism, and glycine degradation (Supp. Fig. S1).

Limma-based DEG analysis revealed 4282 upregulated DEGs and 3078 downregulated DEGs (adj. *P* < 0.05) in SAH liver tissue relative to healthy control liver tissue (Fig. 1c). Notably, the M3 module member *ABHD10* was found to be downregulated in SAH liver tissue. We conducted a linear regression analysis on all M3 DEGs (Supp. Fig. S2) to determine which ones show significant correlations with *ABHD10*. A total of 376 M3 DEGs showed significant correlations with *ABHD10* (adj. *P* < 0.05), with 55 M3 DEGs showing strong correlations ($|r| > 0.5$) with *ABHD10* (Fig. 1d, e). Reactome gene-set enrichment analysis on the *ABHD10*-correlating M3 DEG set ($n = 376$) revealed significant enrichment for peroxisomal protein import (Fig. 1f). Consistently, the ABHD10 enzyme negatively regulates the antioxidant activity of the peroxisomal protein peroxiredoxin 5 (PRDX5) through S-depalmitoylation[8]. As PRDX5 is a key suppressor of oxidative stress in liver tissue[12], ABHD10 downregulation may play a role in suppressing PRDX5-based antioxidant activity and thereby promoting hepatic fibrosis.

### ABHD10 downregulation and S-palmitoylated PRDX5 upregulation associated with ALD in humans and mice.

In order to validate these findings, a series of studies was performed in hepatic tissue samples from an independent AH patient cohort ($n = 10$) and demographically matched healthy control cohort ($n = 10$) (Fig. 2a). We confirmed the presence of several indicators for liver disease in AH patients: reduced serum albumin, enhanced serum total bilirubin, and enhanced INR (Fig. 2b–d). We confirmed that *ABHD10* mRNA expression was markedly downregulated in hepatic tissue samples from AH patients (Fig. 2e). Moreover, significant correlations were observed between *ABHD10* mRNA expression and serum albumin ($r = 0.73$, $p < 0.05$), serum total bilirubin ($r = -0.72$, $p < 0.05$), and INR ($r = -0.70$, $p < 0.05$) (Fig. 2f–h). By immunohistochemical staining analysis, ABHD10 protein was expressed at substantially lower levels in hepatocytes from AH patients (Fig. 2i). Moreover, ABHD10 protein expression was downregulated, while S-palmitoylated Prdx5 was upregulated, in hepatic tissue samples from AH patients (Fig. 2j).

Human cirrhotic liver tissue and CCl4-induced murine fibrotic liver tissue display upregulation of the pro-fibrogenic transcription factor ELK-3 (alternatively NET, ERP, SAP-2)[7]. Our in silico promoter analysis revealed one conserved ELK-3 binding site in the *ABHD10* promoter region 128-138 bp upstream of the TSS (Fig. 2k). AH was associated with an increase in ELK-3 expression (Fig. 2l). As opposed to *ABHD10*, three other ELK-3 targets *EGR1*[7], *FOS*[13], and *HIF1A*[14] were markedly upregulated in hepatic tissue samples from AH patients (Fig. 2m).

In order to validate our human findings, a series of studies was performed in murine models of early- or advanced-stage ALD (Fig. 3a). The early-stage ALD model (i.e., EtOH for 3 weeks under HFD conditions) displayed liver damage and hepatocyte steatosis without substantial fibrosis (Fig. 3b–h). The severe ALD murine model system (i.e., CCl4 for nine weeks followed by EtOH under HFD conditions) exhibited more pronounced hepatic damage with pericellular fibrosis (Fig. 3b–h). Oxidative stress in liver tissue can be assayed through activity levels of the oxidative stress markers MDA, MPO, and NO[15]. The severe ALD murine model system displayed higher hepatic oxidative stress levels as indicated by enhanced activity levels of MDA, MPO, and NO relative to the early-stage ALD model (Supp. Fig. S3a). We also

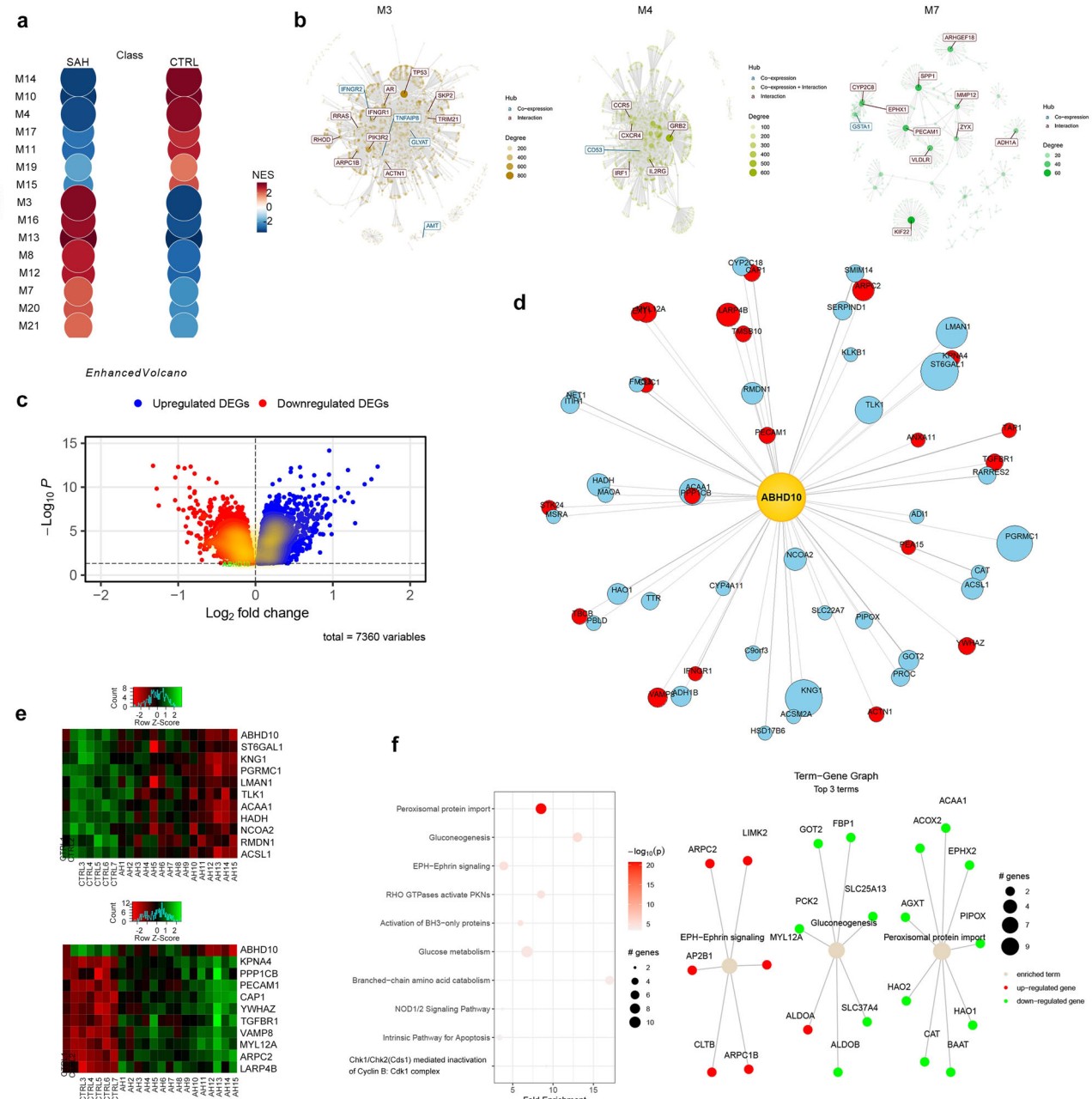

**Fig. 1 Bioinformatics analysis identifies _ABHD10_ as a downregulated gene in SAH patients. a** Co-expression module identification analysis identifies differential co-expression gene modules from severe alcoholic hepatitis (SAH) patients ($n = 15$) versus healthy control (Ctrl) ($n = 7$) liver tissue samples (GEO acc. no.: GSE28619). The size of the circle is proportional to the module gene membership, while the color corresponds to the normalized enrichment score (NES). **b** Network diagrams for the three largest differential co-expression gene modules (M3, M4, and M7), highlighting their hub genes. **c** Volcano plot showing significantly upregulated and downregulated M3 differentially-expressed genes (DEGs) in SAH versus healthy control liver tissue samples (adj. $p < 0.05$). **d** Star diagram illustrating M3 DEGs positively-correlating ($r > 0.5$, FDR <0.05; blue) or negatively-correlating ($r < -0.5$, FDR <0.05; red) with _ABHD10_ (gold hub). The node size is proportional to the $|r|$ value, while the node's distance from the gold hub is proportional to the FDR. **e** Heatmaps illustrating the top ten most positively-correlating M3 DEGs (top) and top ten most negatively-correlating M3 DEGs (bottom) with _ABHD10_. **f** Reactome pathway enrichment analyses for the _ABHD10_-correlating M3 DEGs.

assayed hepatic activity levels of endogenous antioxidant enzymes, including CAT, GPx, GST, and SOD[15]. The severe ALD murine model system displayed lower hepatic activity levels of CAT, GPx, GST, and SOD relative to the early-stage ALD model (Supp. Fig. S3b).

Abhd10 mRNA and protein expression were downregulated, while S-palmitoylated Prdx5 was upregulated, in hepatic tissue samples from both ALD models, with the severe ALD model displaying more profound changes (Fig. 3i, j). Consistently, the expression of Elk-3 and the Elk-3 targets _Egr1_, _Fos_, and _Hif1a_ were enhanced in both ALD models, with the severe ALD model displaying more profound changes (Fig. 3k, l). Therefore, Abhd10 downregulation and S-palmitoylated Prdx5 upregulation are associated with ALD in humans and mice.

**ABHD10 inhibits ROS generation and promotes mature hepatocyte function by downregulating S-palmitoylated PRDX5.** We next explored whether decreases in hepatocyte

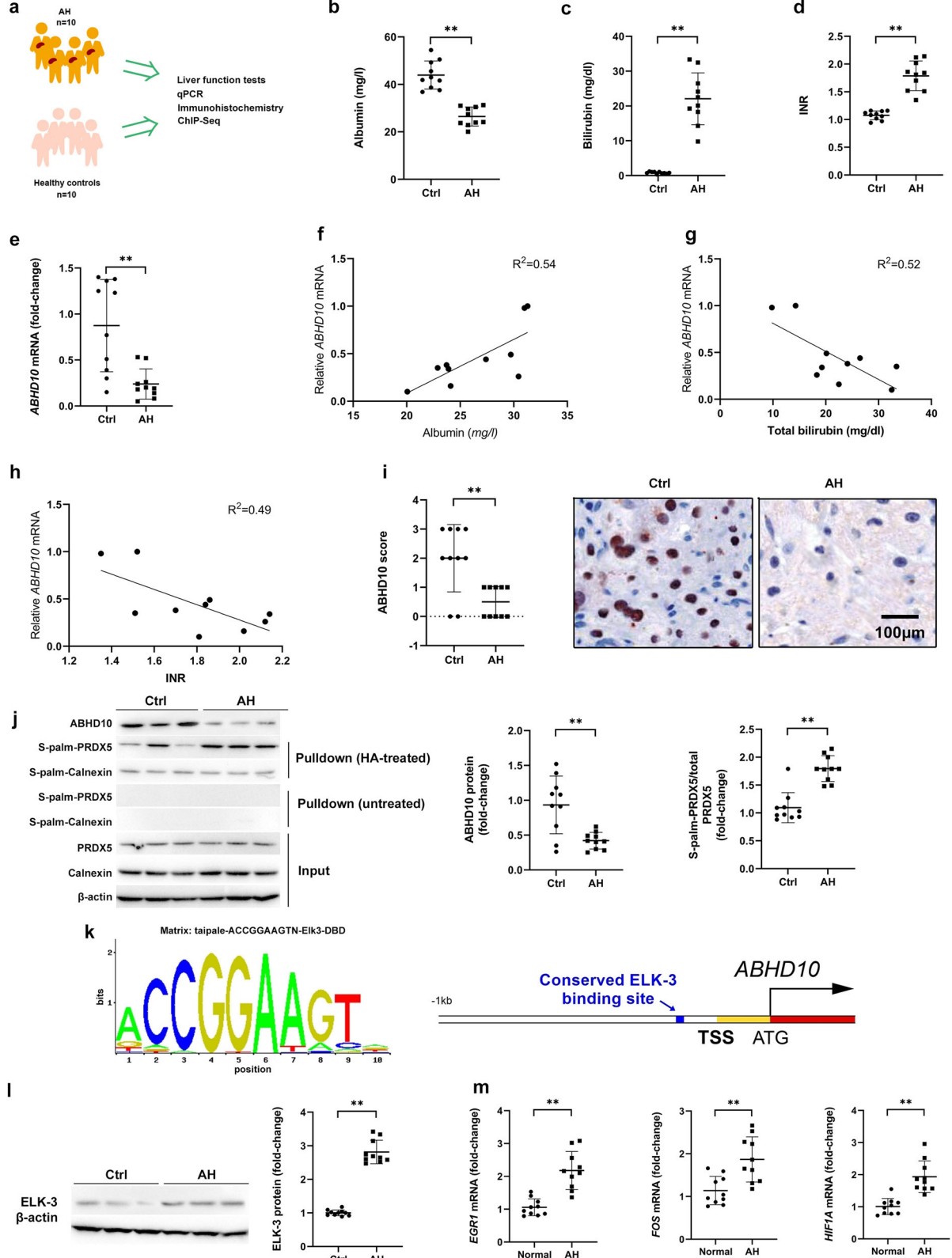

ABHD10 expression are linked to reductions in key biological activities associated with mature hepatocytes, including synthetic activity, metabolic functionality, and bile acid homeostasis. To test this, we utilized an extensively studied model system in which human tumor-derived hepatocyte-like cells undergo dedifferentiation[16]. Initially, 2% DMSO was used to treat human tumor-derived HepaRG cells for two weeks to differentiate them into HepaRG-tdHep cells. These cells were then grown under de-differentiating conditions (i.e., low confluence without DMSO) (Fig. 4a). As expected, de-differentiation resulted in upregulation of EMT markers and progenitor cell-related gene markers (Supp. Fig. S4). Notably, de-differentiation of these hepatocytes was associated with a rapid downregulation in ABHD10 mRNA and protein expression as well as upregulation of S-palmitoylated

**Fig. 2 ABHD10 downregulation and S-palmitoylated PRDX5 upregulation associated with human ALD. a** Blood and liver tissue samples from healthy control (Ctrl) individuals ($n = 10$) and severe alcoholic hepatitis (SAH) patients ($n = 10$) were analyzed. **b** Serum albumin, **c** bilirubin, and **d** INR in SAH patients and healthy controls. **e** *ABHD10* levels in SAH patients and healthy controls analyzed via qPCR. *HPRT1* was used as a housekeeping control. **f–h** Correlations between relative *ABHD10* levels and **f** serum albumin, **g** bilirubin, and **h** INR levels in SAH patients. **i** ABHD10 protein expression was semi-quantitatively scored (0, no/low; 1, medium; 2, high; 3, very high) in SAH and control patients using ABHD10-specific antibodies. Scale bar = 100 µm. **j** ABHD10 and S-palm-PRDX5 protein levels respectively densitometrically measured via standard immunoblotting and streptavidin pulldown-based immunoblotting. Calnexin was used as an S-palmitoylated control, and β-actin was used as a loading control. **k** The Taipale ELK-3 DNA-binding motif (left) and schematic of the *ABHD10* promoter region ($-1$ kb from the transcription start site [TSS]) with the conserved Taipale ELK-3 binding site marked in blue (right). **l** ELK-3 protein levels densitometrically measured via immunoblotting. β-actin was used as a loading control. **m** Expression of the ELK-3 targets *EGR1*, *FOS*, and *HIF1A* analyzed via qPCR. *HPRT1* was used as a housekeeping control. Data presented as medians with IQRs. *$P < 0.05$, **$P < 0.01$ [**b–d**, **e**, **j**, **l**, **m** $U$-test; **f**, **g** Pearson correlation test; and **i** Fisher's exact test].

PRDX5 (Fig. 4b, c). Consistently, the expression of Elk-3 (Fig. 4d) and the Elk-3 targets *EGR1*, *FOS*, and *HIF1A* (Supp. Fig. S5) were enhanced by de-differentiation.

To assess the functional role of ABHD10, a series of hepatocyte function studies were conducted in Hep3B and PHH cells following modulation of ABHD10 expression. ABHD10 over-expression in Hep3B cells (Supp. Fig. S6) was associated with S-palmitoylated PRDX5 downregulation (Fig. 4e). Conversely, ABHD10 knockdown in PHH cells led to increases in S-palmitoylated PRDX5, which was rescued by ectopic over-expression of WT PRDX5 but was unaffected by ectopic overexpression of non-palmitoyable, catalytically-inactive PRDX5 mutant PRDX5$^{C100S8}$ (Fig. 4f and Supp. Fig. S7a, b). Given the known role of the ABHD10-PRDX5 axis in regulating ROS generation[8], ABHD10 knockdown upregulated ROS production in PHH cells, which was rescued by ectopic overexpression of WT PRDX5 but not PRDX5$^{C100S}$ (Fig. 4g). Moreover, ABHD10 knockdown downregulated synthetic, secretory, and metabolic function-related genes (e.g., *ALB*, *BSEP*, *CYP7A1*, *CYP27A1*, *F7*, and *PCK1*) in PHH cells, which were all rescued by ectopic overexpression of WT PRDX5 but not PRDX5$^{C100S}$ (Fig. 4h). ABHD10 knockdown also decreased bile acid synthesis, glycochenodeoxycholate-conjugated bile acid formation, and glucose production in PHH cells, which were all rescued by ectopic overexpression of WT PRDX5 but not PRDX5$^{C100S}$ (Fig. 4i–k). Together, ABHD10 inhibits oxidative stress and improves mature hepatocyte-related biological function by downregulating S-palmitoylated PRDX5.

**TGFβ1 and EGF signaling downregulate hepatocyte ABHD10 via ELK-3.** Next, we sought to evaluate the upstream mechanism(s) governing the dysregulation of ABHD10 and S-palmitoylated PRDX5 in the context of AH-related liver failure. Previously published transcriptomic analyses of AH patient liver samples has highlighted transforming growth factor β1 (TGFβ1) and epidermal growth factor (EGF) as two key upstream regulators of AH progression[17]. Accordingly, we found that AH patient liver samples exhibited the upregulation of TGFβ1, its two primary receptors (1 and 2), and the EGFR ligand amphiregulin (AREG) (Supp. Fig. S8a–d). We posited that TGFβ1 and/or AREG may regulate ABHD10 and S-palmitoylated PRDX5 expression within hepatocytes. Indeed, we found that ABHD10 mRNA and protein levels decreased, while S-palmitoylated PRDX5 increased, following TGFβ1 or AREG treatment in Hep3B cells (Fig. 5a, b). Next, we sought to establish whether the impact of TGFβ1 or AREG treatment on S-palmitoylated PRDX5 upregulation was truly driven by ABHD10 downregulation. When an ABHD10 overexpression vector was transfected into Hep3B cells (Supp. Fig. S9), ABHD10 overexpression abrogated TGFβ1 or AREG-induced S-palmitoylated PRDX5 upregulation (Fig. 5c). Moreover, the impact of TGFβ1 and AREG on ABHD10 were respectively found to be TGFβ1R1-dependent (Fig. 5d) and

EGFR-dependent (Fig. 5e). MEK/ERK signaling was necessary for ABHD10 dysregulation following TGFβ1 or AREG (Fig. 5f, g).

ELK-3 has been shown to be a downstream mediator of TGFβ1 and the RAS/MAPK signaling[7]. Here, ELK-3 knockdown disrupted the ability of TGFβ1 or AREG to promote ABHD10 mRNA downregulation (Fig. 5h, i). Employing the aforedescribed conserved ELK-3 binding site on the *ABHD10* promoter, antibodies specific for ELK-3 were then utilized to perform ChIP. Following treatment with TGFβ1 or AREG, we observed stronger binding of ELK-3 to the region containing the conserved ELK-3 site (Fig. 5j). TGFβ1 or AREG can therefore induce ELK-3 recruitment to the *ABHD10* promoter region. Overall, these data indicate that ABHD10 downregulation and S-palmitoylated PRDX5 upregulation is associated with TGFβ1 or AREG signaling via ELK-3.

Given that PPARγ inhibits TGFβ1 signaling[18], we next assessed the ability of the PPARγ agonist rosiglitazone to antagonize TGFβ1-induced ABHD10 downregulation. Rosiglitazone rescued ABHD10 mRNA downregulation under TGFβ1 conditions (Supp. Fig. S10a). Moreover, rosiglitazone exhibited a dose-dependent impact on *ABHD10* mRNA expression (Supp. Fig. S10b). As rosiglitazone can counteract TGFβ1-induced ABHD10 downregulation, this may provide one mechanism by which PPARγ agonists exhibit beneficial activity in experimental models of ALD[19].

**Ectopic Abhd10 overexpression ameliorates liver damage and oxidative stress in ALD model mice.** In order to determine the therapeutic effects of Abhd10 in vivo, we performed rAAV-based delivery of the *Abhd10* gene to our early-stage and advanced-stage ALD murine models (Fig. 6a). Abhd10 mRNA and protein expression were overexpressed, while S-palmitoylated Prdx5 was downregulated, following rAAV.Abhd10 delivery in both ALD models (Fig. 6b). As expected, the severe ALD murine model system (i.e., CCl$_4$ for nine weeks followed by EtOH under HFD conditions) exhibited more pronounced hepatic damage and oxidative stress compared to the early-stage ALD model (i.e., EtOH for three weeks under HFD conditions) (Fig. 6c–j and Supp. Fig. S11a, b). Liver damage, hepatocyte steatosis, and oxidative stress in the early-stage ALD model were ameliorated by rAAV.Abhd10 delivery (Fig. 6c–j and Supp. Fig. S11a, b). Moreover, the hepatic damage, pericellular fibrosis, and oxidative stress in the severe ALD murine model system were also ameliorated by rAAV.Abhd10 delivery (Fig. 6c–j and Supp. Fig. S11a, b). Given the positive effects of rAAV.Abhd10 delivery on ameliorating alcohol-induced liver damage in mice, ABHD10 may show promise as a viable target for therapeutic intervention in ALD.

## Discussion

Hepatotoxic injury is characterized by cellular and metabolic changes that drive oxidative stress[20,21]. For example, ethanol metabolism within liver cells consumes nicotinamide adenine

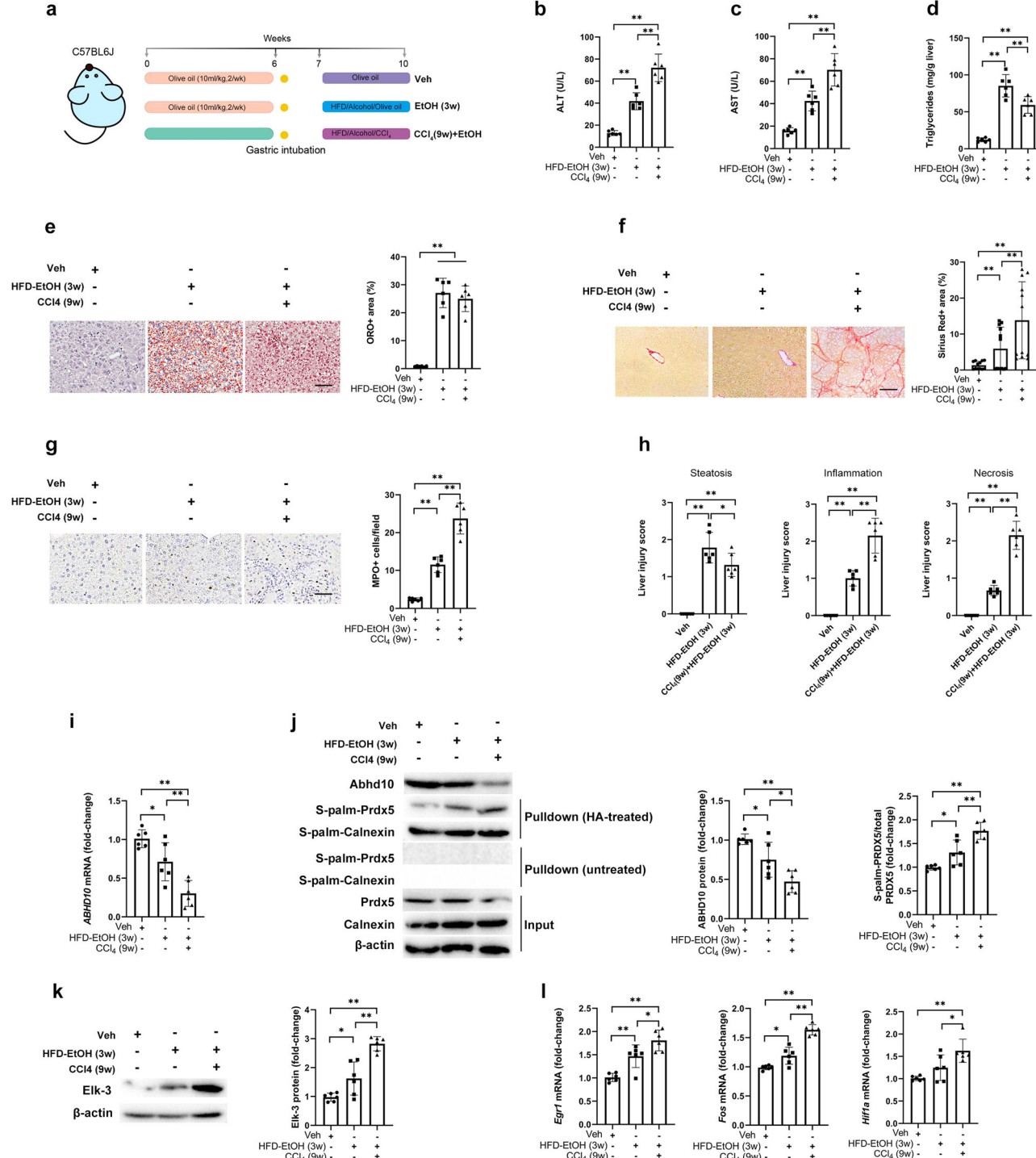

**Fig. 3 ABHD10 downregulation and S-palmitoylated PRDX5 upregulation associated with murine ALD. a** Male C57BL/6 J mice were randomly assigned to three experimental groups (*n* = 6/group): (i) the vehicle group treated for 6 weeks with olive oil, intubation, 1-week rest, and 3 weeks with olive oil; (ii) the EtOH (3w) group treated for 6 weeks with olive oil, intubation, one-week rest, and 3 final weeks with intragastric EtOH; and (iii) the CCl₄ (9w)+EtOH group treated for 6 weeks with CCl₄ (0.2 ml/kg), intubation, 1-week rest, and 3 weeks with a lower CCl₄ dose (0.1 ml/kg) and intragastric EtOH. Mice were sacrificed for experimental analyses. **b**, **c** Assessment of serum transaminase (ALT and AST) levels. **d–f** Hepatic steatosis was quantified by analyzing **d** hepatic triglyceride levels as well as **e** Oil Red O (ORO), and **f** Sirius Red staining in five random 200× fields of view. Scale bar = 200 μm. **g** Myeloperoxidase (MPO)-positive cells quantified from five random 200× fields of view. Scale bar = 200 μm. **h** Assessment of composite liver injury scores. **i** Liver tissue RNA was used for qPCR analyses of *ABHD10*. *HPRT1* was used as a housekeeping control. **j** ABHD10 and S-palm-PRDX5 protein levels respectively densitometrically measured via standard immunoblotting and streptavidin pulldown-based immunoblotting of liver tissue lysates. Calnexin was used as an S-palmitoylated control, and β-actin was used as a loading control. **k** Elk-3 protein levels densitometrically measured via immunoblotting of liver tissue lysates. β-actin was used as a loading control. **l** Liver tissue RNA were used for qPCR analyses of the Elk-3 targets *EGR1*, *FOS*, and *HIF1A*. *HPRT1* was used as a housekeeping control. Data presented as means with SDs. **P* < 0.05, ***P* < 0.01 [one-way ANOVA].

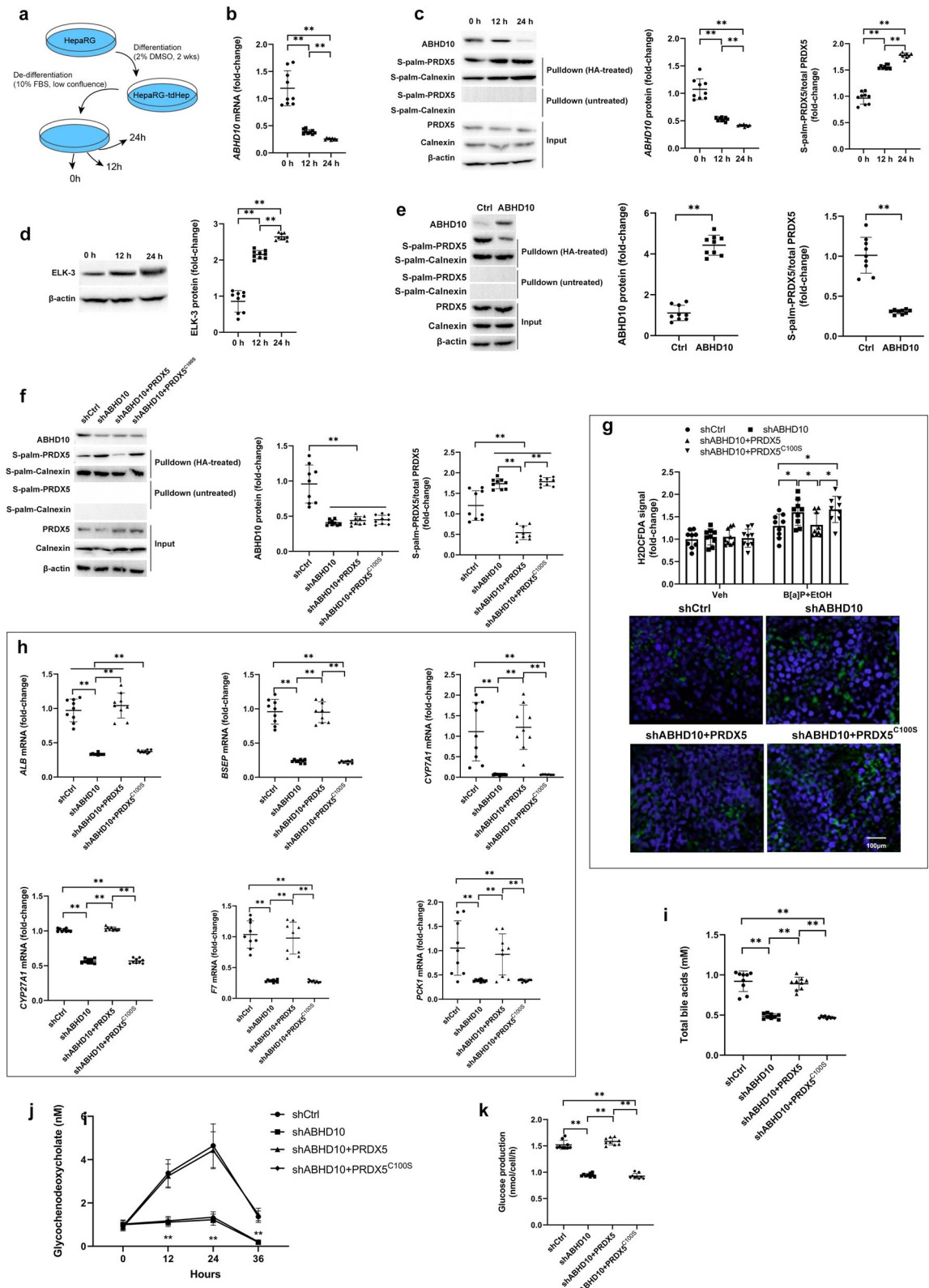

dinucleotide (NAD$^+$), thereby increasing the NADH/NAD$^+$ ratio and driving mitochondrial ROS generation, including singlet oxygen, superoxide (O2$^{·-}$), hydroxyethyl radicals, and ethoxy radicals[22]. As excessive ROS is destructive to liver cells, controlling oxidative stress levels ameliorates liver disease in ALD models[23,24] and other animal models of hepatotoxic injury[25,26]. Peroxiredoxins (PRDXs, PRXs) are a large family of thiol-

dependent peroxidases that reduce mtROS through their conserved cysteine residues[27]. Among the six known mammalian PRDX isoforms (PRDX1-6), PRDX5 is a atypical 2-cysteine (Cys100/Cys204) PRDX with a well-known cytoprotective role, broad substrate specificity (e.g., H$_2$O$_2$, alkyl hydroperoxides, and peroxynitrite), and wide subcellular distribution[28] (e.g., mitochondria, cytoplasm, and nucleus). Accordingly, adenoviral

**Fig. 4 ABHD10 inhibits ROS generation and promotes mature hepatocyte function by downregulating S-palmitoylated PRDX5. a** After differentiation of HepaRG cells into HepaRG-tdHep cells via 2% DMSO, HepaRG-tdHep cells were subjected to de-differentiating conditions. Cells were collected at 0 h (baseline), 12 h, and 24 h time points during de-differentiation ($n = 9$ biological replicates/group) for analyses. **b** *ABHD10* levels analyzed via qPCR. *HPRT1* was used as a housekeeping control. **c** ABHD10 and S-palm-PRDX5 protein levels respectively densitometrically measured via standard immunoblotting and streptavidin pulldown-based immunoblotting. Calnexin was used as an S-palmitoylated control, and β-actin was used as a loading control. **d** ELK-3 protein levels densitometrically measured via immunoblotting. β-actin was used as a loading control. **e** A plasmid encoding the *ABHD10* cDNA or empty control (Ctrl) plasmid were transfected into Hep3B cells ($n = 9$ biological replicates/group). At 48 h after transfection, cells were collected to assess ABHD10 and S-palm-PRDX5 protein levels. Calnexin was used as an S-palmitoylated control, and β-actin was used as a loading control. **f–k** Plasmids encoding the *ABHD10* shRNA (shABHD10), *ABHD10* shRNA and *PRDX5* cDNA (shABHD10 + PRDX5), *ABHD10* shRNA and *PRDX5* C100S cDNA (shABHD10 + PRDX5^C100S), or empty control (Ctrl) plasmid were transfected into primary human hepatocytes (PHH) ($n = 9$ biological replicates/group). **f** Assessment of ABHD10 and S-palm-PRDX5 protein levels 48 h after transfection. Calnexin was used as an S-palmitoylated control, and β-actin was used as a loading control. **g** Transfected PHH cells were treated with vehicle or 25 mM EtOH+2.5 μM B[a]P for 14 days. ROS production was assessed via the fluorescent H2DCFDA dye. Scale bar = 100 μm. **h** Expression of liver metabolism and bile acid transport genes (*ALB, BSEP, CYP7A1, CYP27A1, F7*, and *PCK1*) were assessed via qPCR 48 h after transfection. *HPRT1* was used as a housekeeping control. **i** Total bile acid levels in supernatants were quantified 48 h after transfection. **j** Mass spectrometry was utilized to evaluate supernatant glycochenodeoxycholate levels were collected at 0 h (baseline), 12, 24, and 36 h after transfection. **k** Glucose production was quantified 48 h after transfection. Data presented as medians with IQRs except for panel (**g**) depicting means with SDs. *$P < 0.05$, **$P < 0.01$ [**b–f**, **h**, **i**, **k** *U*-test; **g** two-way ANOVA; and **j** repeated measures ANOVA].

Prdx5 overexpression in mice has been shown to reduce oxidative stress-induced liver injury (e.g., hepatocyte ballooning, endothelial cell injury, and sinusoidal congestion) and improve overall survival in small-for-size liver graft recipients[29]. This evidence suggests that improving hepatocyte PRDX5 activity can have beneficial effects on hepatotoxicity.

PRDX activity is regulated by a variety of post-translation modifications, including hyperoxidation, cysteine glutathionylation, cysteine nitrosylation, tyrosine/threonine phosphorylation, and lysine acetylation[27]. S-palmitoylation, the reversible addition of a saturated $C_{16}$ lipid palmitate moiety to cysteine's sulfur atom via a thioester bond, is the most common form of cysteine acylation in mammals[30–32]. S-palmitoylation impacts the stability, activity, localization, and trafficking of the target protein[33–35], and S-palmitoylation appears to play a role in numerous disease states[36–38]. S-palmitoylation is dynamically regulated by the S-palmitoylators protein acyltransferases and the de-S-palmitoylators acyl protein thioesterases. Although most acyl protein thioesterases are primarily localized in the cytoplasm, lysosomes, and Golgi apparatus, Cao et al. discovered that the acyl protein thioesterase ABHD10 exclusively localizes to the mitochondria and promotes PRDX5 activity through de-S-palmitoylation of its active site Cys100 [8]. Consistent with Cao et al.'s biochemical model, we found that ABHD10 inhibits oxidative stress and promotes mature hepatocyte function via PRDX5 de-S-palmitoylation on its Cys100 residue. Moreover, ABHD10 transactivation and consequent PRDX5 de-S-palmitoylation were negatively regulated by the TGFβ1 and EGF signaling pathways that have been previously associated with hepatotoxic injury and fibrosis[39]. This is consistent with previous research showing growth factor-mediated regulation of other acyl protein thioesterases[40]. Applying this knowledge in vivo, we also demonstrated that ectopic Abhd10 overexpression ameliorates liver damage in ALD model mice. This combined evidence suggests that enhancing hepatocyte ABHD10 expression, and consequent PRDX5 de-S-palmitoylation, can have beneficial effects in hepatotoxic injury and fibrosis.

In conclusion, liver tissue from SAH patients and ALD mice exhibit upregulation of ELK-3 expression and ELK-3 signaling activity coupled with downregulation of ABHD10 and upregulation of deactivating S-palmitoylation of PRDX5. Moreover, TGFβ1 and AREG were found to induce ABHD10 downregulation and S-palmitoylated PRDX5 upregulation in hepatocytes via ELK-3, ultimately driving oxidative stress and disrupting mature hepatocyte function. In vivo, ectopic Abhd10 overexpression ameliorated liver damage in ALD model mice. This evidence suggests that efforts to target the ABHD10-PRDX5 axis may be of value for the treatment of ALD and other forms of hepatotoxicity.

## Methods

**Ethics statement.** All human protocols were approved in advance by the Ethics Committee of the Affiliated Hospital of Chifeng University (approval no. fsyy202005; Chifeng, China) and were performed in accordance with the Declaration of Helsinki guidelines. Written informed consent was obtained from all human participants prior to tissue sample collection.

All animal protocols were approved in advance by the Ethics Committee of the College of Basic Medical Science at Chifeng University (approval no. jcyxy202008; Chifeng, China) and were compliant with the National Institutes of Health (NIH) Guidelines for the Care and Use of Laboratory Animals (Bethesda, MD).

**Bioinformatics analysis.** Affymetrix Human Genome U133 Plus 2.0 microarray data pertaining to human liver tissue samples from SAH ($n = 15$) and normal livers ($n = 7$) were downloaded from the GEO database (GEO acc. no.: GSE28619). The clinical characteristics have been detailed in the source study[11]. The Co-Expression Modules Identification Tool (CEMiTool) package in R utilizes an inverse gamma distribution-based unsupervised filtering method to select genes from the source gene expression file[41]. A soft-thresholding power β = 9 was employed to determine a similarity criterion between gene pairs, and genes were separated into co-expression gene modules using the Dynamic Tree Cut package. Gene-set enrichment analysis was employed to visualize the gene modules that are induced or repressed in the SAH and normal phenotypes. Module network graphs displaying interacting genes were based on the BioGRID *Homo sapiens* interaction file. A Reactome-based over-representation analysis via the clusterProfiler R package was employed to identify and rank pathway enrichment for each module. The limma package in R was employed to assess differential gene expression[42]. Linear regression analyses on DEGs were performed using the lm function in R. Reactome enrichment analyses using the pathfindR package in R[43] was employed to identify and rank pathway enrichment for significantly-correlating DEGs. The promoter region for the *ABHD10* gene (NM_018394) (i.e., 1000 bp upstream from the transcription start site [TSS chr3:111697827]) was searched for four key ELK-3 transcription factor motifs (i.e., M01982 [TRANSFAC20113], ACCGGAAGTN-Elk3-DBD [taipale], M6208_1.02, and MA0759.1 [JASPAR_CORE_2016]) with Contra v3[44] tool (stringency parameters: core = 0.95, similarity matrix = 0.85).

**Collection and analysis of tissue specimens from human donors.** Whole blood and liver biopsy samples were collected from AH patients ($n = 10$) and demographically matched healthy control individuals ($n = 10$) at the Affiliated Hospital of Chifeng University (Chifeng, China). The clinicodemographic characteristics of these human tissue donors are detailed in Supp. Table S1. All AH patients possessed clinically, histologically confirmed diagnosis of AH and were subjected to liver biopsy prior to any treatment. Healthy control donors possessed no history of alcohol abuse, and all healthy control liver specimens were histologically confirmed to be disease-free. Individuals with a history of malignancy, non-alcoholic fatty liver disease (NAFLD) by Kleiner's criteria[45], or HCV infection were excluded from candidacy. Neither sex nor gender were considered in the study design.

Blood samples were subjected to serum albumin, serum total bilirubin, and international normalized ratio (INR) analyses using conventional protocols. Liver tissue samples were subjected to qPCR, immunohistochemical staining, and Western blotting analysis as described below.

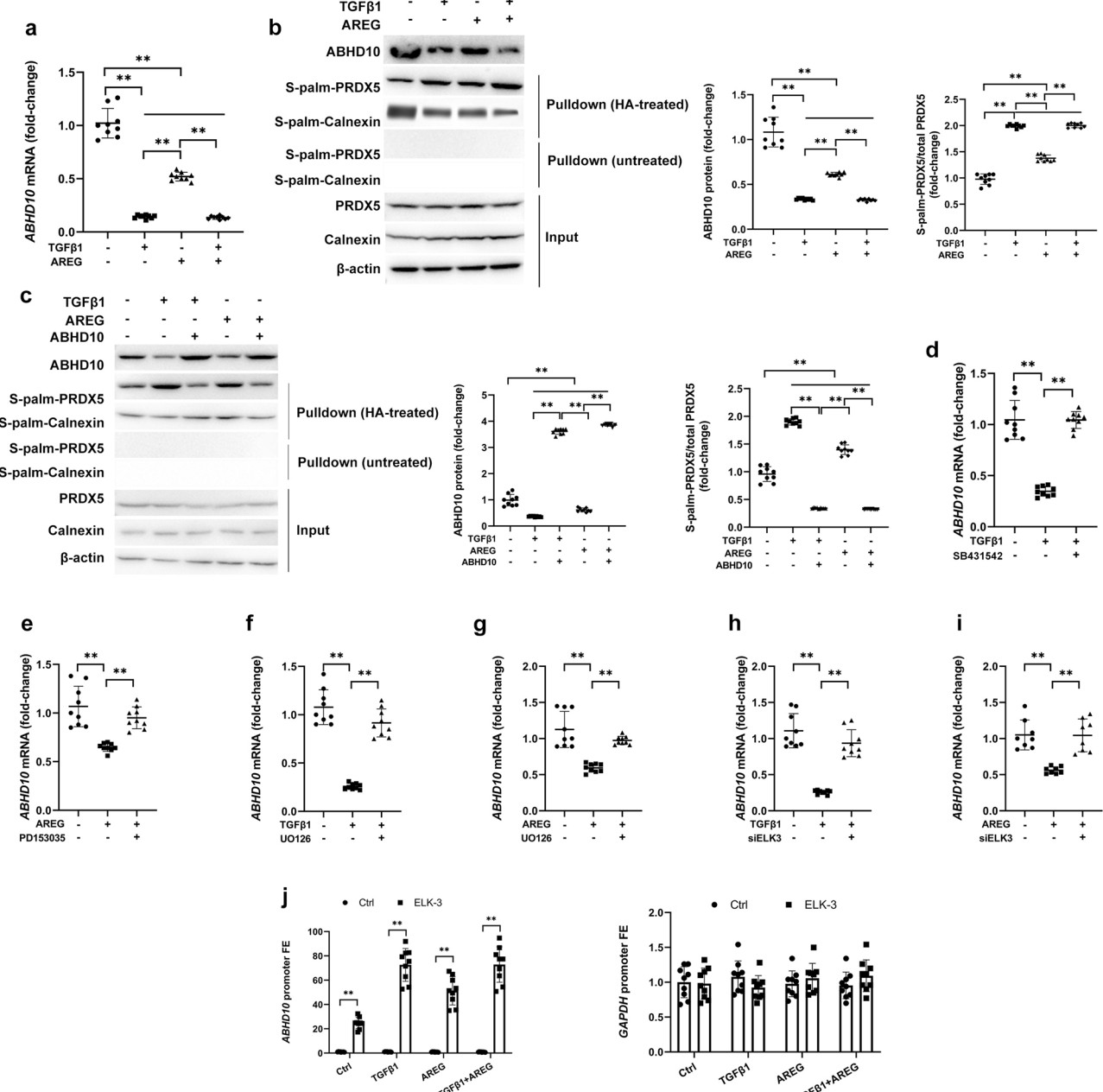

**Fig. 5 TGFβ1 and EGF signaling downregulate hepatocyte ABHD10 via ELK-3.** Following serum-starvation overnight, Hep3B cells were subjected to TGFβ1 (5 ng/ml) for 8 h and/or AREG (50 nM) for 12 h where indicated ($n = 9$ biological replicates/group). **a**, **b** Hep3B cells were exposed to TGFβ1 and/or AREG. **a** ABHD10 levels analyzed via qPCR. HPRT1 was used as a housekeeping control. **b** ABHD10 and S-palm-PRDX5 protein levels respectively densitometrically measured via standard immunoblotting and streptavidin pulldown-based immunoblotting. Calnexin was used as an S-palmitoylated control, and β-actin was used as a loading control. **c** At 48 h following transfection with a plasmid encoding the ABHD10 cDNA or empty control (Ctrl) plasmid and treatment with TGFβ1, ABHD10, and S-palm-PRDX5 protein levels were respectively densitometrically measured via standard immunoblotting and streptavidin pulldown-based immunoblotting. Calnexin was used as an S-palmitoylated control, and β-actin was used as a loading control. ABHD10 expression was assessed via qPCR following: **d** pretreatment with SB431542 (5 nM) to inhibit TGFβ-RI and treatment with TGFβ1; **e** pretreatment for 12 h with PD153035 (25 μM) to inhibit EGFR and treatment with AREG; **f**, **g** pretreatment with U0126 (10 μM) to inhibit MEK/ERK activity and treatment with **f** TGFβ1 or **g** AREG; and **h**, **i** pretreatment with siELK3 to knockdown ELK-3 and treatment with **h** TGFβ1 or **i** AREG. HPRT1 was used as a housekeeping control. **j** Following TGFβ1 and/or AREG treatment, ChIP was performed using control IgG or a ChIP-grade antibody specific for ELK-3. The ABHD10 promoter region (left) and GAPDH promoter region (right) were assessed via qPCR, with data being shown in the form of ELK-3 enrichment relative to control IgG. Data presented as medians with IQRs except for panel (**j**) that depicts means with SDs. *$P < 0.05$, **$P < 0.01$ [**a**–**i** $U$-test and **j** two-way ANOVA].

**In vivo models of ALD**. Mice were housed in a climate-controlled facility (12 h light/dark cycle) with free food and water access. Male C57BL/6 mice (6 weeks old) were obtained from the Beijing Vital River Laboratory Animal Technology Co. (Beijing, China). Acute or chronic hepatic injury models were established in male C57BL/6 mice (12 weeks old, 20–25 g) as reported previously[46]. Olive oil (vehicle control; AGRIC, Greece), CCl₄ (purity >99.5%, Sigma), and 190-proof (95%) EtOH

were utilized to induce liver injury as described in a prior study[17]. Briefly, mice were randomized by random number generator into three experimental cohorts: vehicle, EtOH (3w), and CCl₄ (9w) + EtOH. Vehicle cohort and EtOH (3w) cohort animals were injected intraperitoneally (i.p.) with olive oil (10 ml/kg), while CCl₄ (9w)+EtOH cohort animals were i.p. injected with CCl₄ (0.2 ml/kg in 10 ml/kg olive oil), twice weekly over a 6-week period. At the end of week 6, all animals were

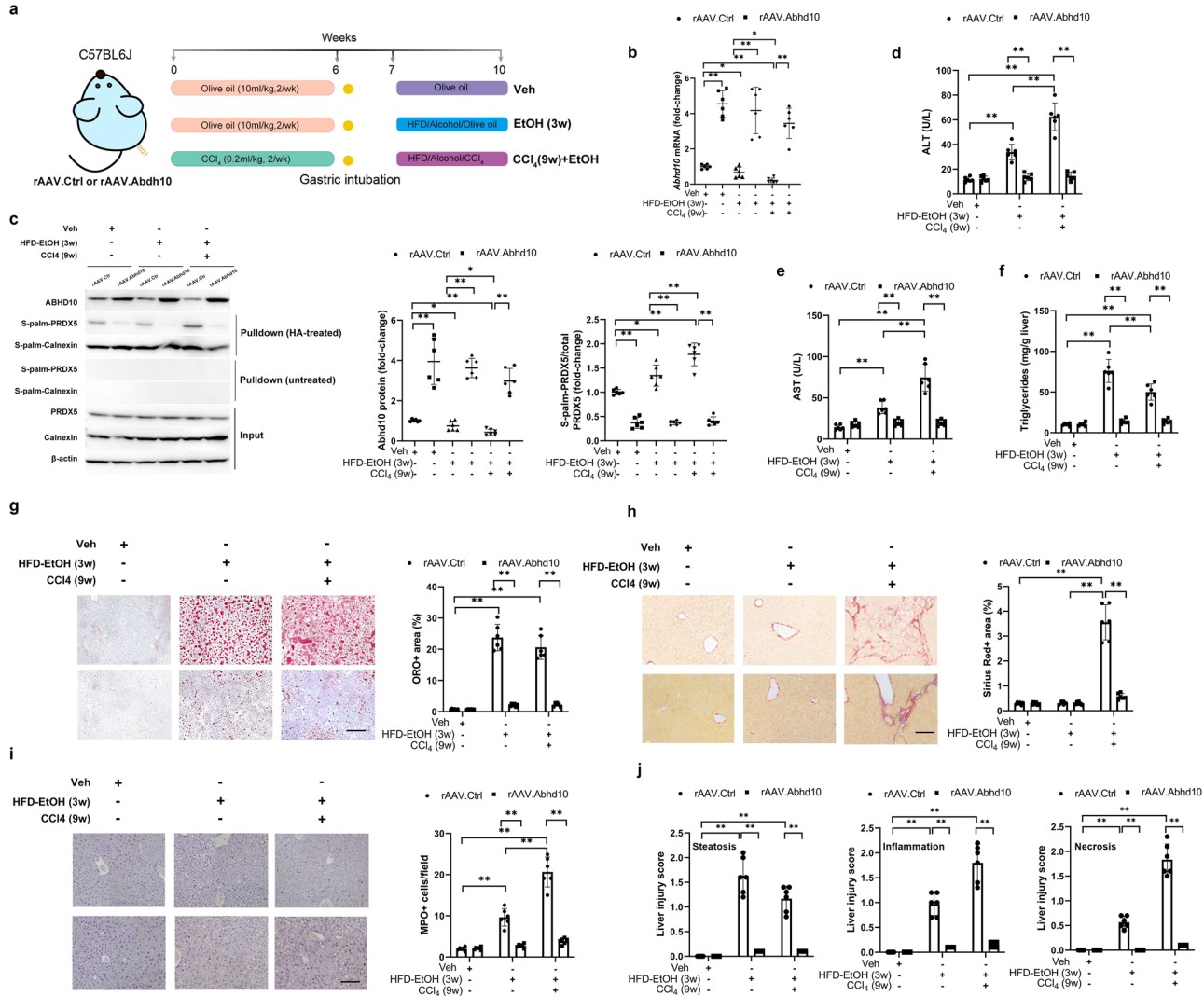

**Fig. 6 ABHD10 downregulation and S-palmitoylated PRDX5 upregulation associated with murine ALD. a** Male C57BL/6 J mice were intraperitoneally injected with rAAV.Ctrl or rAAV.Abhd10. Mice were then randomly assigned into six experimental groups (n = 6/group): (i, ii) the vehicle groups treated for 6 weeks with olive oil, intubation, 1-week rest, and 3 weeks with olive oil; (iii, iv) the EtOH (3w) groups treated for 6 weeks with olive oil, intubation, 1-week rest, and 3 final weeks with intragastric EtOH; and (v, vi) the CCl₄ (9w)+EtOH groups treated for 6 weeks with CCl₄ (0.2 ml/kg), intubation, 1-week rest, and 3 weeks with a lower CCl₄ dose (0.1 ml/kg) and intragastric EtOH. Mice were sacrificed for experimental analyses. **b** Liver tissue RNA was used for qPCR analyses of *ABHD10*. *HPRT1* was used as a housekeeping control. **c** ABHD10 and S-palm-PRDX5 protein levels respectively densitometrically measured via standard immunoblotting and streptavidin pulldown-based immunoblotting of liver tissue lysates. Calnexin was used as an S-palmitoylated control, and β-actin was used as a loading control. **d, e** Assessment of serum transaminase (ALT and AST) levels. **f–h** Hepatic steatosis was quantified by analyzing **f** hepatic triglyceride levels as well as **g** Oil Red O (ORO) and **h** Sirius Red staining. Scale bar = 200 μm. **i** Myeloperoxidase (MPO)-positive cells quantified from five random 200× fields of view. Scale bar = 200 μm. **j** Assessment of composite liver injury scores. Data presented as means with SDs except for panels (**b, c**) that depicts medians with IQRs. *P < 0.05, **P < 0.01 [**b, c** U-test and **d–j** two-way ANOVA].

intragastrically intubated, individually housed in metabolic cages, and given a one-week recovery period with free food and water access. Then, for the final three weeks, all animals were allowed to freely consume water and non-nutritious cellulose pellets. Vehicle cohort and EtOH (3w) cohort animals were injected intraperitoneally (i.p.) with olive oil (10 ml/kg), while CCl₄ (9w)+EtOH cohort were i.p. injected with CCl₄ (0.1 ml/kg in 10 ml/kg olive oil), twice weekly over the 3-week period. During the same final 3-week period, the EtOH (3w) cohort and CCl₄ (9w)+EtOH cohort animals were administered a high-fat diet (HFD) supplemented with EtOH, with alcohol being steadily administered (16 g/kg/d) via an intragastric cannula initially with this rate being gradually increased to 25 g/kg/d. Tolerance development was assessed by evaluating alcohol intoxication. At the study endpoint, pentobarbital (50 mg/kg, i.p.) was used to anesthetize mice and collect blood. After collection, liquid nitrogen was used to snap-freeze tissues.

**Construction and delivery of recombinant adeno-associated virus (rAAV) vectors**. The enhanced green fluorescent protein (eGFP) cassette was excised from an rAAV2 plasmid under the control of the liver-specific promoter 1 (rAAV2-

pLSP1-eGFP) to obtain the rAAV2-pLSP1 vector backbone[47]. The murine *Abdh10* cDNA (NM_172511; #MC203394, Origene) was PCR-amplified and cloned into the rAAV2-pLSP1 backbone to form rAAV.Abhd10, while the empty rAAV2-pLSP1 vector was used as a negative control (rAAV.Ctrl). As recombinant AAV type 8 (rAAV8) vectors show the most pronounced liver tropism in mice among rAAV vectors[48], the rAAV.Ctrl or rAAV.Abhd10 vectors were pseudoserotyped with rAAV8 by calcium phosphate transfection with pXX680 and p5E18-VD2/8. The rAAV.Ctrl or rAAV.Abhd10 particles were produced in HEK293 cells and then purified and titrated[48]. For hepatocyte-specific gene delivery, mice were intraperitoneally injected with rAAV.Ctrl or rAAV.Abhd10.

**Biochemical analyses**. Serum levels of aspartate aminotransferase (AST), alanine aminotransferase (ALT), albumin, and total bilirubin were analyzed with their respective biochemical kits (Jiancheng Institute of Biotechnology, Nanjing, China). Hepatic triglyceride levels were assessed using a triglyceride assay kit (Solarbio, Beijing, China). Myeloperoxidase (MPO) activity levels in liver homogenate samples were assessed with an ELISA kit (Jiancheng Institute of Biotechnology).

Malondialdehyde (MDA) and nitric oxide (NO) levels in liver homogenate samples were assessed with their respective colorimetric kits (Jiancheng Institute of Biotechnology). Antioxidant marker levels in liver homogenate samples (namely, catalase (CAT), glutathione peroxidase (GPx), glutathione-*S*-transferase (GST), and superoxide dismutase (SOD)) were assayed with their respective kits (Cayman Chemicals Co.).

**Hepatic histopathology and immunohistochemical staining**. Sirius Red staining was performed by de-paraffinization of 5-μm sections followed by staining with Sirius Red as well as hematoxylin and eosin. Oil Red O staining was performed by embedding tissues with optimal cutting temperature compound and preparing 10-μm tissue sections. ImageJ software (NIH) was used to quantify Oil Red O and Sirius Red staining results, with ten random 100× magnification fields of view per sample being analyzed. Hematoxylin and eosin was utilized to stain deparaffinized 3-μm sections based on standard staining protocols. Two licensed pathologists analyzed hepatic histopathology in a blinded manner[49].

Immunohistochemical staining was performed by deparaffinizing and rehydrating 3-μm sections followed by microwave-heating for 40 min in Tris-EDTA (pH 6.0) for antigen retrieval. Following peroxidase blocking using REAL (Dako) for 10 min, sections were exposed to an anti-ABHD10 antibody (1:20, #PA5-57905, Thermo) for 60 min. Antibody staining was developed using the REAL EnVision Detection System (Dako, Japan). Hematoxylin was then used for counterstaining, and Immu-Mount (Thermo) was used to mount samples. Two licensed pathologists reviewed and scored all stained samples in a blinded manner. A semi-quantitative approach, based on a scale of 0–3, was used to score the immunohistochemical staining.

**Cell culture**. Primary human hepatocytes (PHH, Lonza) were prepared and cultured in media purchased from Lonza, including thawing media (MCHT), plating media (MP), and maintenance media (MM), which were utilized for the thawing, plating, and culturing of these cells, respectively. After being allowed to attach for 4 h to collagen-coated 6- or 12-well plates (Corning), Matrigel (0.3 mg/ml; Corning) was applied to the culture surface. For small-interfering RNA (siRNA)-based knockdown experiments, Matrigel overlaying was performed at 6 h post-siRNA transfection, with cells being cultured in low-serum OptiMEM (Gibco) during that 6-h period. Cells and supernatants were collected at the indicated times.

Mycoplasma-free Hep3B cells (ATCC) were cultured in DMEM (Gibco) containing 10% FBS (Gibco) and penicillin/streptomycin (Gibco). For experiments involving drug treatment, cells were subjected to serum-starvation by culturing them for 2 h in DMEM containing 1% FBS prior to drug exposure. For knockdown and overexpression assays, treatment was initiated 48 h post-transfection, with cells being cultured in OptiMEM for the initial 6 h post-transfection period followed by culture in DMEM containing 1% FBS until subsequent collection.

Mycoplasma-free HepaRG cells (Thermo), an immortalized hepatic cell line, were cultured in Williams E media containing 10% FBS, insulin (5 μg/mL), hydrocortisone hemisuccinate (50 μM), and penicillin/streptomycin. Following an initial 2-week culture period, 2% DMSO was added to this culture media, and cells were cultured for 2 more weeks. For analyses of hepatocyte dedifferentiation, HepaRG cells were harvested at plated at a low density without any DMSO in the culture media[50].

**Gene silencing and overexpression**. The siRNAs were as follows: Silencer Select Negative Control No. 1 siRNA (#4390843, Thermo), Silencer Select siRNA against *ABHD10* (#4392420, Thermo), and Silencer Select siRNA against *ELK3* (#4390771, Thermo). For PHH cells and Hep3B cells, siRNAs were used at respective working concentrations of 20 pM and 10 pM. Lipofectamine-RNAiMAX (Invitrogen) was used to transfect siRNAs into cells. Experiments were conducted at 48 h post-transfection.

For ABHD10 overexpression, the human *ABHD10* clone (NM_018394, SC319244, Origene) was cloned into a pcDNA6 vector (Invitrogen) under the control of a CMV promoter. For ABHD10 knockdown with or without rescue overexpression of WT *PRDX5* or non-palmitoyable, catalytically-inactive PRDX5 mutant PRDX5^C100S, a modified pcDNA6 vector containing an *ABHD10* shRNA (under control of a U6 promoter) combined with an overexpression cassette containing a scrambled control cDNA, *PRDX5* cDNA, or the *PRDX5*^C100S cDNA (under control of a CMV promoter and an internal ribosomal entry site) was constructed[51]. Lipofectamine 3000 (Invitrogen) was used to transfect plasmids. Experiments were conducted at 48 h post-transfection.

**Measurement of reactive oxygen species (ROS) generation**. ROS generation was assayed using B[a]P and EtOH as previously described in ref. [52]. Briefly, PHH cells were cultured for 2 days in MM and were then treated every two days with B[a]P (2.5 μM in MM) plus EtOH (25 mM in MM) or vehicle for 14 days. Twenty-four hours after the last B[a]P+EtOH treatment, ROS generation was assessed using the H2DCFDA dye to detect hydrogen peroxide generation using a SpectraMax M2 plate reader (Molecular Devices) with 485/520 nm excitation/emission wavelengths. Fluorescence intensity values were normalized to total protein content. An inverted epifluorescence microscope was used to capture image samples.

**Measurement of biliary acid, glycochenodeoxycholate, and glucose production**. PHH cells were added ($2 \times 10^4$/well) to 96-well collagen-coated plates in MM (Lonza), transfected with the indicated siRNAs, and collected after 48 h. A Total Bile Acid Assay Kit (Cell Biolabs) was used to measure biliary acid content using a SpectraMax M2 plate reader (Molecular Devices). A calibration curve was used to measure total bile acid levels in individual samples.

Glycochenodeoxycholate levels were quantified via mass spectroscopy[53]. Briefly, 50 μl of cultured PHH cells and 100 μl of acetonitrile were vortexed, centrifuged for 2.5 min at $22,0000 \times g$, and supernatants were diluted tenfold in a solution of acetonitrile in water (1:4), followed by LC-MS/MS analysis. A standard curve (1–10,000 nM) was prepared with glycochenodeoxycholate (Sigma), with linear regression analyses being used to quantify glycochenodeoxycholate concentrations in the media.

PHH cells were added ($1 \times 10^5$/well) to 12-well collagen-coated plates in MM for 24 h, after which they were subjected to serum-starvation overnight in DMEM containing glucose (1 g/l), L-glutamine (4 mM), and sodium bicarbonate (3.7 g/l). Samples were then incubated in 300 μl of glucose-production medium, which was prepared by combining DMEM with sodium bicarbonate (3.7 g/l), HEPES (15 mM, ThermoFisher): lactate (20 mM, Sigma), glutamine (2 mM), pyruvate (2 mM, Fisher), and pCPT-cAMP (0.1 mM, Sigma) (67,68). Following a 24 h incubation, glucose levels in 50 μl of media were assessed with a Glucose Colorimetric Detection kit (EIAGLUC, Invitrogen). Cells were washed thoroughly before culture such that the cultured hepatocytes were the only potential source of glucose in these supernatants.

**Cellular stimulation and inhibition experiments**. For appropriate experiments, immediately prior to Matrigel application, cells were treated with amphiregulin (AREG, 50 nM, Sigma-Aldrich) or TGFβ1 (5 ng/mL, R&D Systems). For the inhibition experiments, cells were cultured for 2 h in DMEM containing 1% FBS prior to addition of the the TGFβRI inhibitor SB431542 (5 nM, Calbiochem-EMD Millipore), the EGFR inhibitor PD153035 (3 μM, Calbiochem-EMD Millipore), the MEK inhibitor UO126 (10 μM, Promega), or rosiglitazone (10 μM, Sigma), followed 45 min later by treatment with AREG (50 nM) or TGFβ1 (5 ng/mL) as indicated.

**qPCR**. The AllPrep DNA/RNA/Protein kit (Qiagen) was used to collect RNA from tissue samples. TRIzol reagent (Invitrogen) was utilized to collect RNA from cells in culture via a phenol/chloroform approach. A Nanodrop instrument (Thermo) was employed to measure RNA quantity and purity, after which RNA (1 μg) was reverse transcribed with a Maxima First Strand cDNA Synthesis Kit for RT-qPCR with dsDNase (Thermo). All qPCR assays were conducted in a 96-well plate-based format, with 50 ng of cDNA per reaction. The SsoAdvanced Universal SYBR Green Supermix (Bio-Rad) and a CFX96 Real-Time PCR instrument (Bio-Rad) were used for all qPCR assays. The primer sequences are listed in Supp. Table S2. The $2^{-\Delta\Delta Ct}$ method was utilized to compare relative gene expression using *HPRT1* for normalization[54].

**Chromatin immunoprecipitation (ChIP)-PCR**. Hep3B cells ($1 \times 10^6$) were added to 150-mm cell culture plates in Dulbecco's Modified Eagle Medium (DMEM) containing 10% fetal bovine serum (FBS). During the final 24 h of culture, the media was supplemented with TGFβ1 (5 ng/ml). Oligonucleotides used to amplify the *ABHD10* and *GAPDH* promoter regions were designed and synthesized by Shanghai Sangon Biotech (China). All ChIP-PCR steps were conducted with an EZ-ChIP kit (Millipore)[17] with the overnight immunoprecipitation step performed with anti-ELK3 (5 μg, #NBP1-83960, Novus) or control rabbit IgG (5 μg, #NBP2-50261, Novus).

**Click reaction and streptavidin pulldown for S-palmitoylation detection**. RIPA buffer (0.1% SDS, 150 mM NaCl, 50 mM Tris pH 7.5, 1% Triton X-100) supplemented with freshly added cOmplete EDTA-free protease inhibitors (Sigma), 40 mM DTT, and phosphatase inhibitors (2 mM NaF, 1 mM Na₃VO₄, 2 mM β-glycerophosphate) was used to lyse cells and tissue samples. Lysates were subjected to a click reaction using biotin azide as previously described in ref. [55]. Briefly, the click reaction was conducted at room temperature for 1 h in the darkness under gentle rotation using 100 μl lysis buffer containing 50 μg lysate protein, 1 mM CuSO₄, 1 mM tris(2-carboxyethyl)phosphine hydrochloride (TCEP), 100 μM biotin azide, and 100 μM tris((1-benzyl-1H-1,2,3-triazol-4-yl)methy-l)amine (TBTA). Reactions were halted with 20 μl 6× SDS-sample loading buffer (50 mM Tris-HCl [pH 6.8], 30 mM EDTA, 48% glycerol, 9% MeSH, 6% SDS, and 0.03% bromphenol blue). For the hydroxylamine (HA)-treated samples, HA (2.5% v/v) was added to the reaction mixture for 30 min prior to the addition of the 6× SDS-sample loading buffer. S-palmitoylation can be inferred by the enrichment of a target protein in the HA-treated samples relative to the untreated control samples[56].

The biotin-tagged proteins were mixed with streptavidin agarose beads (Life Technologies) for 3 h at room temperature under rotation. The streptavidin agarose beads bound with tagged proteins were then washed thrice with immunoprecipitation buffer (150 mM NaCl, 50 mM Tris-HCl (pH 7.4), 5 mM EDTA, and 0.5% NP40). The bound proteins were eluted with elution buffer (95%

formamide with 10 mM EDTA (pH 8.2)) at 95 °C for 10 min. Samples were subjected to Western blotting as described below.

**Western blotting**. Cells and tissue samples were lysed as described above. Liver tissue samples were diluted 1:20 (mg:µl) prior to sonication (50 W probe, 20% amplitude; five cycles of 20 s/cycle). Western blotting was performed by diluting 40 ug of protein per sample in Laemli buffer (AlfaAesar), heating these samples for 3 min at 95 °C, and separating them via SDS-PAGE (Bio-Rad) prior to transfer onto a nitrocellulose membrane (pore size: 0.2 µm, Bio-Rad). Blots were subsequently blocked using 5% non-fat milk in TBS-T for 1 h, followed by incubation overnight with appropriate primary antibodies (Supp. Table S3). β-actin was used as a loading control. Blots were then washed thrice with TBS-T and probed for 1 h with species-appropriate HRP-conjugated secondary antibodies (Supp. Table S3). Finally, a chemiluminescence substrate (ECL, Bio-Rad) was added to the membranes, and blots were resolved with ChemiDoc (Bio-Rad). Band quantification as quantified using ImageJ software.

**Statistics and reproducibility**. All statistical analyses were performed using GraphPad Prism 7. Bar and line charts display means ± standard deviations (SDs). Box-and-whisker plots display medians, interquartile ranges (IQRs, 25th–75th percentiles), and individual values. All gene and protein expression analyses were normalized to the median or mean values of the control samples. Comparisons between groups were assessed using unpaired two-tailed Student's $t$-test, Fisher's exact test, two-tailed Mann–Whitney $U$-test, one-way ANOVA, repeated measures ANOVA, or two-way ANOVA as indicated in the Figure legends. No statistical methodology was used to predetermine experimental sample sizes. All in vivo experiments were performed with $n = 6$ mice per cohort, while all in vitro experiments were performed with $n = 9$ independent biological replicates per group. Experiments were not randomized with the exception of rAAV studies, where mice were block randomized to receive rAAV.Ctrl or rAAV.Abhd10. No data were excluded from the reported results; all biological replicates were used for statistical analyses. Investigators were blinded to group allocation during data analysis.

**Reporting summary**. Further information on research design is available in the Nature Portfolio Reporting Summary linked to this article.

## Data availability

All data generated or analysed during this study are included in this published article and its supplementary information files. Numerical source data for all graphs and charts have been provided in the Supplementary Data file. Uncropped Western blot images have been provided in Supp. Fig. S12.

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

## Acknowledgements

This work was supported by the National Natural Science Foundation of China (grant no. 82060118), the Research Program of Science and Technology at the Universities of Inner Mongolia Autonomous Region (grant no. NJZY20203), and the Program for Young Talents of Chifeng University (grant no. CFXYYT2202).

## Author contributions

Conceived and designed the study: T.-Z.L. and C.-Y.B. Performed the experimental procedures: T.-Z.L., C.-Y.B., B.W., C.-Y.Z., and W.-T.W. Analyzed the data: SIR-IGULENG, T.-W.S., and J.Z. Drafted the manuscript: T.-Z.L.

## Competing interests

The authors declare no competing interests.
