## [Peer Review File · Communications Biology]

Reviewers' comments:

Reviewer #1 (Remarks to the Author):

This manuscript demonstrates a novel connection between ABHD10 expression via ELK-3, PRDX5 regulation via S-palmitoylation, and hepatotoxic injury in ALD. Overall, the conclusions are very intriguing, and add an exciting layer of knowledge in an important therapeutic area, with even some translational potential of ABHD10 expression. However, the data presented is inadequate for publication in its current form. Much of the conclusions are based on measurement of S-palmitoylation of key proteins, namely PRDX5. However, missing controls, incomplete methods, and poor data quality make it impossible to ascertain the likelihood that the results are being interpreted correctly. I think if higher quality data, with proper controls (as set by the current state of the field) were included, and the results held up, this could be a very important piece of work.

Specific issues/suggestions:

- The Methods for detection of S-palmitoylated protein are incomplete. It is completely unclear what was done (presumably this is metabolic labeling with an alkyne lipid, since the methods discuss click reaction, but nothing is described). Therefore, these key results are interpretable.
- Regardless of the experimental details, +/- HA controls are needed for all S-palmitoylation detection blots, especially Fig 2J, 3I, etc
- I don't not understand the results in Fig 4f. Why is expression of active-site mutant PRDX5 increasing the fraction of S-palm PRDX5? I do not see why this should rescue the effect.
- Check for typos (several, especially when writing "S-palmitoylated"

Reviewer #2 (Remarks to the Author):

Authors demonstrate that Elk-3 controls hepatotoxic injury and fibrosis in alcoholic liver disease by directly binding with Abhd10. Throughout the study, there are many concerns and lacks significant results to warrant its publication in Communications Biology.

1. The letters of the analyzed figures are too small to recognize them.
2. In Figure 1, how about the pathways in other modules? Why did authors highlight amino acid metabolism in the text? Authors need to show the reasons for choosing M3 should be clearer. Furthermore, authors need to show that ABHD10 has clearly decreased in ALD as mentioned.
3. In Figure 2L, authors need to show the difference in expression of western blot
4. In Figure 3, authors must mark HFD label in figure.
5. In Figure 3J, 4D and 4E, authors should change the representative blots to one that clearly shows a difference
6. In Figure 4G, the labels overlapped. Authors need to fix the label (shABHD10+PRDX5 into (shABHD10+PRDX5C100s)
7. In Figure 4G, authors need to confirm the significant effect between shABHD10+PRDX5 and shABHD10+PRDX5 mutant for supporting the effect of PRDX5 residue.
8. In Figure 6, did the mice fed with HFD? If so, authors need to describe in the text and figure. Moreover, authors only showed the results of ORO, SIRIUS, MPO and Liver injury graphs in this figure. Authors need to show the images of histological staining for supporting their analysis.
9. In Figure 6, authors need to show the difference between EtOH for 3weeks and CCl4 for 9 weeks with EtOH which is the more severe form as authors described.
10. Authors demonstrated the axis of Elk-3-ABHD10 and PRDX5 regulates ROS production which increased liver damage, however, the related results of ROS production or anti-oxidant effect are too scarce. Authors should show more related results in this article. Moreover, compared to what authors emphasized of the effect of Elk3, the related results are too little to prove this axis both in cells and mice models.

Reviewer 1:

Responses to Reviewers

Q1: The Methods for detection of S-palmitoylated protein are incomplete. It is completely unclear what was done (presumably this is metabolic labeling with an alkyne lipid, since the methods discuss click reaction, but nothing is described). Therefore, these key results are interpretable.

Response: As requested, we have now clearly detailed the Methods for detection of S-palmitoylated protein. Please see the Methods subsection entitled “Click reaction and streptavidin pulldown for S-palmitoylation detection”.

Q2: Regardless of the experimental details, +/- HA controls are needed for all S-palmitoylation detection blots, especially Fig 2J, 3I, etc.

Response: As recommended, we have now added +/- HA controls to all S-palmitoylation detection blots.

Q3: I do not understand the results in Fig 4f. Why is expression of active-site mutant PRDX5 increasing the fraction of S-palm PRDX5? I do not see why this should rescue the effect.

Response: This was a typographical error in the text of the Results. We have corrected the language.

Q4: Check for typos (several, especially when writing “S-palmitoylated”).

Response: We have reviewed the text and corrected the typos.

Reviewer 2:

Q1: The letters of the analyzed figures are too small to recognize them.

Response: Thank you for your constructive feedback in improving our work. We have re-constructed Figure 1 with larger fonts to improve its readability.

Q2A: In Figure 1, how about the pathways in other modules? Authors need to show the reasons for choosing M3 should be clearer.

Response: The Cemi analysis revealed 15 gene modules that were significantly enriched for SAH (adj. p -value <0.05 ; Fig. 1A, Supp. Table S3). M3, M4, and M7 were the largest of these 15 modules (Fig. 1B). We selected the largest module among these -- M3 -- for further investigation. We have provided this rationale in the revised Results.

Q2B: Why did authors highlight amino acid metabolism in the text?

Response: We have now expanded the Results text to include the top three-ranking Reactome pathways for the M3 module.

Q2C: Furthermore, authors need to show that ABHD10 has clearly decreased in ALD as mentioned.

Response: We have provided this data in the Figures. Please see the following panels:

--ABHD10 mRNA expression is markedly downregulated in hepatic tissue samples from AH patients (Fig. 2E).

--ABHD10 protein expression is downregulated in hepatic tissue samples from AH patients (Fig. 2J).

-- Abhd10 mRNA and protein expression are downregulated in hepatic tissue samples from murine models of early- or advanced-stage ALD, with the advanced-stage ALD model displaying more profound changes (Fig. 3H, I).

Q3: In Figure 2L, authors need to show the difference in expression of Western blot.

Response: As requested, we have replaced the Western blots in Figure 2L with more representative blots. The densitometric quantification is provided next to the Western blot.

Q4: In Figure 3, authors must mark HFD label in figure.

Response: As suggested, we have added the HFD label to all Figure 3 panels.

Q5: In Figure 3J, 4D and 4E, authors should change the representative blots to one that clearly shows a difference.

Response: We have replaced the blots in Figures 3J, 4D, and 4E to more representative ones.

Q6: In Figure 4G, the labels overlapped. Authors need to fix the label (shABHD10+PRDX5 into (shABHD10+PRDX5C100s)

Response: We have corrected the label in Figure 4G.

Q7: In Figure 4G, authors need to confirm the significant effect between shABHD10+PRDX5 and shABHD10+PRDX5 mutant for supporting the effect of PRDX5 residue.

Response: We have re-performed this experiment and confirmed the significant effect between

shABHD10+PRDX5 and shABHD10+PRDX5 mutant.

Q8: *In Figure 6, did the mice fed with HFD? If so, authors need to describe in the text and figure. Moreover, authors only showed the results of ORO, SIRIUS, MPO, and liver injury graphs in this figure. Authors need to show the images of histological staining for supporting their analysis.*

Response: As requested, we have added the HFD label to the text and figure. We have added the supporting histological staining images to Figure 6.

Q9: *In Figure 6, authors need to show the difference between EtOH for 3 weeks and CCl4 for 9 weeks with EtOH which is the more severe form as authors described.*

Response: As requested, we have added text to the Results and significance symbols to the Figure 6 panels showing the significant differences between the two models (EtOH for 3 weeks vs. CCl4 for 9 weeks with EtOH).

Q10: *Authors demonstrated the axis of Elk-3-ABHD10 and PRDX5 regulates ROS production which increased liver damage, however, the related results of ROS production or anti-oxidant effect are too scarce. Authors should show more related results in this article. Moreover, compared to what authors emphasized of the effect of Elk3, the related results are too little to prove this axis both in cells and mice models.*

Response: We have added additional data regarding oxidative stress markers and anti-oxidant enzyme levels to the murine models (Supp. Figs. S4, S12). This additional data demonstrates that ectopic Abhd10 overexpression reduces oxidative stress and improves anti-oxidant enzyme

levels in ALD model mice (Supp. Fig. S12).

REVIEWERS' COMMENTS:

Reviewer #1 (Remarks to the Author):

Thank you for addressing my questions and comments. This is a much-improved manuscript.

Reviewer #2 (Remarks to the Author):

The authors answered the questions raised by the reviewers. I think now the manuscript can be published.